# Removal of Contaminants from Water by Membrane Filtration: A Review

**DOI:** 10.3390/membranes12060570

**Published:** 2022-05-30

**Authors:** Jaime Cevallos-Mendoza, Célia G. Amorim, Joan Manuel Rodríguez-Díaz, Maria da Conceição B. S. M. Montenegro

**Affiliations:** 1LAQV-REQUIMTE/Departamento de Ciências Químicas, Faculdade de Farmácia, Universidade do Porto, R. Jorge Viterbo Ferreira, 228, 4050-313 Porto, Portugal; jaime.c.m93@gmail.com; 2Instituto de Admisión y Nivelación, Universidad Técnica de Manabí, Portoviejo 130105, Ecuador; 3Laboratorio de Análisis Químicos y Biotecnológicos, Instituto de Investigación, Universidad Técnica de Manabí, Portoviejo 130105, Ecuador; 4Departamento de Procesos Químicos, Facultad de Ciencias Matemáticas, Físicas y Químicas, Universidad Técnica de Manabí, Portoviejo 130105, Ecuador

**Keywords:** polymeric membranes, water treatment, polymeric additive, nanomaterials, membrane separation, nanostructured membranes, pollutant compounds

## Abstract

Drinking water sources are increasingly subject to various types of contamination due to anthropogenic factors and require proper treatment to remove disease-causing agents. Public drinking water systems use different treatment methods to provide safe and quality drinking water to populations. However, they are ineffective in removing contaminants that are considered a danger to the environment and therefore to humans. Several alternative treatment processes have been proposed, such as membrane filtration, as final purification methods. This paper aims to summarize the type of pollutant compounds, filtration processes, and membranes that have been most studied in this area with particular emphasis on how the modification of membranes, either the manufacturing process or the incorporation of nanomaterials, influences their performance.

## 1. Introduction

Water is considered a “universal solvent”; one that can dissolve a wide variety of molecules due to its molecular structure and properties. Its importance in biological processes makes water such an essential asset for human life that is being continually threatened by climate change and daily sources of contamination [1]. The global warming that we have been assisting is expected to have an impact on contaminant release due to changes in solubility, dissolution kinetics, contaminant gas phase production, sorption equilibrium, biological degradation, and non-aqueous phase liquid mobilization. Anthropogenic activities have largely contributed to the degradation of water quality, affecting rivers, lakes, and oceans around the world, degrading not only the environment but also human health and the communities of living beings that depend on it. The unsustainable development of all nations is putting pressure on water resources, with global demand for water expected to increase by 50% in the next few years [2]. The presence of organic and inorganic pollutants in water resources and their relationship to emerging diseases motivates the growing search for more efficient treatment processes to provide the world’s population with access to safe drinking water [3]. These concerns began in the 17th century when water filters for domestic applications made of wool, sponge, and charcoal were originally used [4]. Guidelines on the quality of drinking water, issued by the World Health Organization, have been published and repealed to protect the population against the dangers that chemical compounds and microorganisms can have on human health. In addition, the Water Framework Directive [5], the EU’s main instrument in the political strategy against water pollution, provides specific measures in this regard. Monitoring of the watch-listed substances is mandatory for all EU member states to establish sustainable measures and strategies to minimize contamination and impact on the aquatic environment. This directive sets minimum quality standards for water intended for human consumption (i.e., drinking, cooking, other domestic purposes), protecting us against contamination.

In search of sustainability, several strategies and different methods of wastewater treatment have been developed; several studies have shown the inefficiency of conventional methods used in wastewater treatment plants (WWTPs) [6,7,8,9]. The conventional treatment methods used in WWTPs include clarification, oxidation, aeration, filtration, and disinfection [9]. Factors such as low volatility, hydrophobic characteristics, pKa, size, shape, charge, and extremely low concentrations may account for inefficient removal of contaminants [10]. Although physicochemical methods have been widely studied for the removal of pollutants in WWTPs, they have disadvantages, namely: (i) high energy costs, (ii) high capital for operation and maintenance, (iii) toxic waste generation, (iv) addition of toxic chemical agents, (v) training of operating personnel, and (vi) low efficiency in the degradation of organic pollutants, among others. Thus, other types of processes such as ion exchange, electrochemical, chemical precipitation, advanced oxidation processes (AOP), and membrane separation have emerged as advantageous alternatives for removing toxic compounds from water. Among the treatment options described, membrane separation processes are of great interest. The high removal rate of low molecular weight pollutants, the ability to integrate with other systems, the possibility of environmental degradation of the materials, the low price, and the ease of regeneration [10] make them very attractive. Membranes emerged as a viable means of water purification in the 1960s with the development of high-performance synthetic membranes, but their application for reverse osmosis was not adopted until the 1980s [4].

Membranes commonly used in filtration processes can be classified into: (a) conventional membranes and (b) commercial composite thin film (TFC) membranes. In the latter, a thin active layer of polyamide (PA) (<200 nm), obtained by interfacial polymerization, is deposited on a porous layer of polyethersulfone (PES) or polysulfone (PSU) (about 50 microns) [11]. Membrane filtration requires a driving force (pressure, concentration, or electrical potential gradients) to separate the desired components that are determined by the pore size of the membrane. The filtration process based on pressure gradients are classified as: microfiltration (MF), ultrafiltration (UF), nanofiltration (NF), reverse osmosis (RO), and forward osmosis (FO) [12]. MF processes are based on the use of membranes with a symmetrical porous structure that allow the separation of particles with an average size greater than 0.1 µm and varying the working pressure between 1 and 3 atm. UF membranes have pore diameters ranging from 0.01 µm to 0.1 µm with slightly higher operating pressures (from 2 to 7 atm); both are often used as a pretreatment step to remove colloids and natural organic matter. NF presents a much lower MW cutoff with an average pore diameter between 1 nm and 10 nm. It has been used to remove divalent salts and other small molecules such as PhACs and emerging micropollutants. RO has been described as the most efficient process in the removal of dissolved inorganic and small organic molecules. However, the reduced size pores (0.1 to 0.6 nm) in the RO membranes requires higher pressures (between 30 and 50 atm) to reverse the natural flow of the water [12]. This condition involves much more energy consumption when compared with NF (5–20 atm), being mentioned as the main disadvantage.

Along with the development of new types of membranes, new nanomaterials have been researched with recognized contributions in several areas including water treatment. Their characteristic surface properties (large specific surface areas and high reactivity) have made them valuable materials to be used as adsorbents, catalysts, and sensors, among other applications [13]. Thus, the latest developments in the field of nanomaterials and nanotechnology have allowed the design of new generations of artificial membranes for water purification with new functions and improved molecular separation properties. These nanomaterial-based membranes, which include nanoparticles, nanofibers, two-dimensional layered materials, and other nanostructured compounds, exhibit extraordinary permeation properties, as well as some additional properties (antifouling, antibacterial, photodegradable, etc.) [14]. Therefore, the interest of the academic community in developing these type of membranes is evidenced by the number of scientific publications in the last 10 years (about 53,500 publications between 2010 and 2021) [15].

Based on the structure of the membrane and how the nanomaterial is dispersed, different types of membranes with different permeation characteristics have been referred as: (a) conventional nanostructured; (b) thin-film nanostructured (TFN); and (c) localized surface nanocomposite [16]. In conventional nanostructured membranes, nanomaterials are incorporated into the polymeric matrix during the phase inversion process, but these membranes have a low tolerance to high temperatures, corrosive environments, and organic solvents [17]. In the last decades, cross-linked polyamide obtained by interfacial polymerization and the TFC membranes derived from it has replaced conventional ones, achieving improvements in separation performance and permeability [18]. TFN membranes are an emerging class of TFC membranes and are formed by incorporating nanoparticles into a thin polyamide layer to modify surface properties, resulting in nanostructured materials with remarkable improvements in performance [18,19]. To our knowledge, there is no systematized review on the advantages and disadvantages of the application of these membranes in the removal of common and emerging pollutants from water. Therefore, this paper aims to show the performance of the described membranes in filtration processes currently used in the purification of water contaminated with microorganisms, toxic metal ions, dyes, and organic and inorganic compounds.

## 2. Membrane Separation Processes

Membrane technology encompasses the scientific and engineering approaches associated with the transport or rejection of components, species, or substances through or by membranes. This technology is widely used in water treatment for domestic and industrial supply, in chemical, biotechnological, pharmaceutical, food, and metallurgical industries, as well as in other separation processes. Figure 1 shows a schematic representation of the most relevant areas where membrane technology has application. Industrial and environmental applications are vast, because membrane separation is a clean technology with reduced energy consumption and it replaces conventional processes such as filtration, distillation, ion exchange, and chemical treatment systems. Furthermore, it allows continuous separation under mild conditions, enables an easy upscaling hybrid processing, and the membrane properties can be adjusted to the expected end. However, this technology has some drawbacks, such as concentration polarization and membrane fouling, low membrane lifetime, and low selectivity and flux. These obstacles can be overcome by designing membranes with different types of morphologies, requiring different biological, chemical, and physical properties according to the type of application [20,21,22,23].

Membranes are generally classified as isotropic or anisotropic. Isotropic membranes are uniform in composition and physical nature across the cross-section of the membrane. Anisotropic membranes are non-uniform over the membrane cross-section and they typically consist of layers that vary in structure and/or chemical composition. The nature of the raw material (organic or inorganic) and the desired morphology (dense or porous) influences the technique choice for membrane preparation. For that, membranes can be prepared by sintering, stretching, clamping tracks, coating solutions, and phase inversion [24], enabling the preparation of many different membrane characteristics, which is considered a promising alternative for contaminant removal by a membrane separation process [24].

The most versatile technique that allows the preparation of all kinds of polymeric membranes is phase inversion [24]. The phase inversion technique is a process whereby a stable polymer solution is transformed from liquid to solid state in a few milliseconds under important thermodynamic and kinetic aspects. Different morphologies can be obtained by controlling several chemicals and/or physical factors. The most widely used technique to produce phase inversion membranes is the immersion precipitation, also known as non-solvent induced phase separation (NIPS) [25]. Membranes formed through this process are often in a flat sheet or hollow fiber configuration. The polymer dissolved in the solvent (polymer solution) is cast on a proper supporting layer (ex. silicon wafer) for a flat sheet configuration or forced by a single orifice spinneret for hollow fibers. After the homogeneous spread of the polymer, the coagulation takes place in a bath containing a nonsolvent. The precipitation (phase separation) takes place during solvent and nonsolvent exchange and can lead to a variety of asymmetric or symmetrical structures [26,27]. The exchange rate between the solvent contained in the cast film and the nonsolvent present in the coagulation bath determines the membrane morphology. Other approaches are considered to optimize the morphology and properties of membranes such as the use of different types of additive-modified polymers and/or nanomaterials, as shown in the following section. Morphology, hydrophilicity, and permeability can be optimized by using polymer additives, because they mainly act as pore-forming agents [28,29]. To improve the selectivity and removal efficiency of pollutants during the filtration process, different specific nanomaterials have been selected, providing higher removal capacity, antimicrobial and photocatalytic activity, improved hydrophilicity, and mechanical resistance, among other features [30]. The next sections describe in more detail the strategies followed in the preparation of the membranes for the most significant contaminants to improve their filtration and removal process.

### 2.1. Pharmaceutical Compound Removal

The widespread use of pharmaceutical active compounds (PhACs) by humans and animals results in the contamination of the aquatic environment with serious repercussions on human health. Despite the most innovative processes used to remove these compounds, their persistence in drinking water is still a reality. Several research works have reported on the development of new targeted strategies for the removal of these compounds from water by membrane filtration [31,32,33]. However, this cleaning process is quite complex because its efficiency is determined by the physicochemical properties of the PhACs, the pH of the solutions, the composition of the membrane, the interactions between the solute and the membrane, and also the simultaneous presence of several chemically related compounds [10]. In the last 20 years, a few scientific research papers (Table 1) were dedicated to the removal of pharmaceutical residues from water by polymeric membranes.

Membrane processes for wastewater reclamation/reuse and drinking water treatment have been evaluated for the remotion of PhACs by microfiltration (MF), nanofiltration (NF), ultrafiltration (UF), and reverse osmosis (RO) and combinations of membranes in series [34]. MF and UF have a limited application for the removal of PhACs in aqueous media due to the larger molecular weight cutoff (MWCO) of the membrane when compared with the molecular weight (MW) of most PhACs (150–500 g/mol) [35]. NF and RO present a much lower MWCO, thereby allowing a high rejection of PhACs. NF and RO separation have been described as the most efficient processes in the removal of PhACs [12]. According to the literature, microfiltration and ultrafiltration are generally not fully effective in removing PhACs [10,36]. Although the literature suggests that the use of methods involving NF and RO are potentially efficient for the removal of pharmaceuticals from wastewater, some authors present unconvincing results. For example, the use of a polyethersulfone (PES) nanofiltration membrane prepared by phase inversion has shown to be not fully efficient in removing carbamazepine, diclofenac, and ibuprofen from drinking water. Vergili [37] studied the rejection of three PhACs through a PES NF membrane. The results showed that the overall rejection was approximately 31–39% and 55–61% for neutral (carbamazepine) and ionic (diclofenac and ibuprofen) PhACs, respectively. Considering that diclofenac and ibuprofen are negatively charged, under the experimental conditions electrostatic repulsion contributed to the better rejection obtained for these PhACs relative to carbamazepine. These findings were confirmed by other authors [38], who have shown that negatively charged compounds are more removed by negative surface charged membranes than neutral and positive ones. However, the low overall removal efficiency can be explained by the smaller molecular size of PhACs relative to the pore size of these membranes [37].

Kimura et al. [39] investigated the rejection of neutral (uncharged) PhACs with different molecular weights by two types of RO membranes: (1) a TFC-PA (polyamide membrane (PA)) and (2) a cellulose acetate (CA) membrane. As shown in Table 1, the former generally exhibits a higher efficiency in removing PhACs, compared with the CA membrane. While for TFC-PA membranes the rejection tendency is dictated by the molecular weight of the compound (size exclusion), for CA membranes it is the polarity that determines the rejection. These results demonstrate that the dominant rejection mechanism for RO membranes depends on the membrane material and the physicochemical properties of the target compounds. These findings confirm the results previously obtained by the same authors [40], who showed that the rejection of non-charged compounds was generally lower (<90% except for one case) and mainly influenced by the molecular size of the compounds; being the negatively charged compounds more effectively rejected (>90%) regardless of other physicochemical properties of the tested compounds due to electrostatic exclusion. Furthermore, they showed that RO membranes always exhibited slightly better rejection compared with NF membranes, probably because the former has a lower MWCO, as confirmed by other authors [41,42] (Table 1). For example, Urtiaga et al. [41] achieved a removal rate greater than 99% for atenolol, gemfibrozil, ibuprofen, and naproxen using TFC-RO membranes, while only 85% was removed with TFC-NF membranes [42]. Therefore, PhAC removal is influenced not only by the membrane composition but also by the type of filtration process used.

The removal performance of TFC membranes for PhACs has also been studied by the incorporation of nanoparticles into/under the polyamide layer membrane [43,44]. Dong et al. [43] studied the rejection of various PhACs through a thin-film nanostructured (TFN) nanofiltration membrane, based on the preparation of an in situ polysulfone support embedded with zeolite nanoparticles followed by interfacial polymerization to form the polyamide layer. This membrane had a similar rejection performance for negatively charged PhACs (>90%) when compared with TFC membranes. Concerning the neutral or positively charged PhACs, the smaller rejection ability was confirmed. However, in the same study it was noticed that even with a similar surface hydrofobicity of the membrane regarding TFC, a much higher water permeability was observed attributed to the internal pores of zeolite nanoparticles, the increased membrane surface roughness, and, though undesirably, the microporous defects between the nanoparticles and the polyamide matrix.

The skin layer structure of the membranes modified with additives and nanosized particles have been improved regarding both permeability and rejection profile, mechanical strength, stiffness, antifouling characteristics of the host polymer, and especially selectivity towards certain compounds [43,44,45,46,47,48]. An increase in water permeability and rejection performance for most of the tested PhACs was achieved due to a higher pore formation efficiency, improved wettability, and a substantial increase in active surface area. Although it is recognized that nanoscience and nanotechnologies offer excellent opportunities for the development of innovative techniques for water treatment, information regarding the modification of membranes with nanomaterials and their efficiency in removing PhACs from aqueous matrices is still scarce. Nevertheless, studies emphasize that the modification of membranes with composite materials (CNTs, nanofibers, zeolites, etc.,) may be an advantage in water treatment processes, making filtration more effective in removing PhACs and reducing membrane fouling and energy consumption, which are currently the major limitations of the process [13]. This was evidenced by the aforementioned work [47], in which the rejection of tetracycline and 17β-estradiol through a conventional UF nanostructured membrane was studied, concluding that the excellent physicochemical properties of mesoporous hollow carbon nanospheres (MHCNs) and their high surface area contributed to a rejection rate of over 90% with ultrafiltration being able to operate at very low pressure. The achieved rejection rate (94%) competed with that presented by Kimura et al. [39] when using polymeric RO membranes (polyamide and cellulose acetate) without any nanosized additive. The authors affirm that the improved removal capacity is mainly due to the high adsorption capacity of the incorporated MHCNs governed by hydrophobic interactions.

Nadour et al. [46] synthesized conventional nanostructured UF membranes using PSU as membrane matrix, methylcellulose (MC) as a pore-forming agent, and commercial powdered activated carbon (PAC) as adsorbent nanomaterial. The addition of activated carbon to the membrane matrix improved the removal of PhACs by 34% and 6% for acetaminophen and diclofenac, respectively, compared with conventional membranes without nanomaterials. Despite this improvement, removal rates are not satisfactory enough, possibly to be improved by using RO instead of UF membranes. According to the authors, the carbon–polymeric membranes can remove trace pharmaceuticals from water due to the hybrid process coupling filtration and adsorption. However, different removal rates were obtained for similar molecular weight of diclofenac (50.4%) [46] and estradiol (94%) [47], that used different carbon nanoparticles in an ultrafiltration membrane with a similar molecular weight cutoff. These results show how nanomaterials can be used to modulate the membrane performance, as already stated.

Kuttiani et al. [48], reported the removal of up to 99.9% of acetaminophen and 87% of ibuprofen from water through an ultrafiltration polyimide membrane embedded with reline-functionalized nanosilica. The membrane removes 84.9% and 76% more acetaminophen and ibuprofen than membranes without any nanoparticles. These results are due to the presence of silanol groups present in the functionalized silica nanoparticles, which are available active sites for PhAC adsorption through the hydrogen bonding mechanism. The greater elimination of acetaminophen compared with ibuprofen is attributed to the greater interaction of the former with nanoparticles in the membrane matrix. Consistently, the acetaminophen molecule has more binding sites (two proton acceptors (NH and C=O)) and one donor group (OH)) than ibuprofen (OH, proton donor, and C=O, proton acceptor). Therefore, acetaminophen could exhibit enhanced interactions with functionalized silica nanoparticles on the membrane surface during filtration.

More recently, Zhou et al. [49] used TiO_2_ to modify a PVDF ultrafiltration membrane for sulfadiazine removal from water. The results showed a high rejection capacity with removal rates of about 91.4%. These results are similar to those previously reported by Dong et al. [43] for a zeolite-incorporated TFN membrane, but with a membrane requiring lower operating pressure. The authors attribute this high removal rate to the photodegradation of sulfadiazine that results from the photocatalytic activity of TiO_2_ [49].

MOFs have recently been reported as promising adsorbents for the removal of PhACs from water, regarding its large surface area and controlled porosity [50], outperforming commercially activated carbon powder [51]. This nanomaterial has been used in ultrafiltration processes referred to as hybrid MOF-UF systems for better results [52]. That combination presents a higher retention rate when compared with simple UF [52]. These investigations confirm the high performance of MOFs as an adsorbent material that makes them a suitable alternative to improve the efficiency of membrane filtration processes in the removal of PhACs. Basu et al. [44] prepared TFN nanofiltration membranes for PhAC removal from water by using different fabrication processes and incorporating a class of MOF known as zeolitic imidazolate framework-8 (ZIF-8). Two different membranes were prepared: (a) polysulfone (PSU) support membrane with ZIF-8 and polyamide (PA) separation layer and (b) layer-by-layer (LBL) polyamide/ZIF-8 nanostructured membrane on top of PSU support. The latter preparation process yields a superior removal rate for acetaminophen as the membrane structure is defect free.

Despite the excellent properties of TFN membranes, the requirement of high operating pressures due to the reduced pore size is still a drawback. To overcome these issues and maintain the high selectivity in the removal of organic and inorganic pollutants, as well as reasonable fluxes, and lower operating pressures, supramolecular structures such as cyclodextrins (CDs) have gained ground in this area [53,54,55,56]. Their great ability is mainly attributed to the capacity to form inclusion complexes with a variety of target molecules, such as organic pollutants due to their cage-like shape and hydrophobic cavity [57]. For example, Wang et al. [58] reported nearly complete removal (about 99.9%) of an antihypertensive (propranolol) and an endocrine disruptive compound (bisphenol A) at ultrahigh water flux, approximately two orders of magnitude higher than commercial nanofiltration membranes with similar rejection functions. The macroporous membranes were dopped with high levels of β-cyclodextrin polymers (β-CDP) and prepared in a conventional way. The excellent adsorption characteristics were due to the synergistic effect of rapid β-CDP adsorption, abundant β-CDP nanoparticles, and the large contact area offered by spongy pores.

Magnetic nanoparticles (Fe_3_O_4_) combined with acid-treated multi-walled carbon nanotubes (MWCNTs) were also proposed for the removal of Bisphenol A (BPA) and Norfloxacin (NOR) from water (Table 1). Ultrafiltration (UF) membranes prepared through a phase inversion process, incorporating COOH-MWCNTs and COOH-MWCNTs/Fe_3_O_4_ nanocomposites into polyvinyl chloride (PVC) were tested [59]. The use of acid-treated MWCNTs or acid-treated MWCNTs with magnetic nanoparticles improved the removal rate of both compounds when compared with pristine membranes. Comparing membranes containing acid-treated MWCNTs with or without magnetic nanoparticles, the former had a superior rejection for BPA, while membranes without magnetic nanoparticles had relatively higher rejection for NOR. The higher removal of BPA was attributed to the increased hydrophilicity of the membrane combining the metal oxide with the MWCNTs, improving anti-fouling property. Regarding NOR, the acid-treated MWCNTs had special adsorption effects on it, while Fe_3_O_4_ inhibited the performance of MWCNTs. Different fillers could be specifically selected for different pollutants. On the other hand, Muhamad et al. [60] reported up to 87% removal of BPA from water through an ultrafiltration PES membrane incorporated with SiO_2_. The results shown in Table 1 demonstrate BPA removal rate extention was much higher for PES membranes containing SiO_2_ compared with other reported membranes [59]. This indicates that SiO_2_ is a more suitable nanomaterial for improving membrane efficiency to remove BPA.

**Table 1 membranes-12-00570-t001:** Application of polymeric membranes in the purification of water contaminated with PhACs.

PhACs Class	Name	Process	Polymer	Additive	Nanomaterial	% Removal	Sample	Ref.
Antibiotics	Sulfadiazine	UF	PVDF	PVP	TiO_2_	91.4	synthetic	[49]
		NF	PA_TF_	PVP	Zeolite	>90	synthetic	[43]
	Amoxicillin	NF	PES	PVP	-	56–99	wastewater	[61]
	Ampicillin Cephalexin-hydrate Ciprofloxacin Erythromycin Nalidixic acid Norfloxacin Roxithromycin Sulfamethazine		PA_TF_	PVP	-	>90	synthetic	[43]
					Zeolite			
	Chloramphenicol	NF	PA_TF_	PVP	-	81	synthetic	[43]
					Zeolite	84		
	Sulfamethoxazole	NF	PA_TF_	PVP	-	>90		[43]
		RO	CA_TF_	-	-	82		[39]
			PA_TF_	-	-	70		[39]
	Tetracycline	UF	PES	PVP	HMCN	97		[47]
Antidepressants	Sulpiride	NF	PA_TF_	PVP	-	>90	synthetic	[43]
					Zeolite			
Antihistamine	Ranitidine	NF	PA_TF_	PVP	-	88	synthetic	
					Zeolite	84		
	Nizatidine	NF	PA_TF_	PVP	-	>90		
					Zeolite			
Anti-hypertensives	Atenolol	NF	PA_TF_	-	-	>85	synthetic	[42]
		RO				99.5		[41]
	Diltiazem	NF	PA_TF_	PVP	-	>90	synthetic	[43]
					Zeolite			
	Metoprolol	NF	PA_TF_	-	-	>85	synthetic	[42]
				PVP	-	88		[43]
					Zeolite	82		
	Propranolol	UF	PVDF	PVP	β-CDP	99.9	synthetic	[58]
	Primidone	NF	PA_TF_	-	-	87	synthetic	[40]
		RO	CA_TF_			85		[39]
			PA_TF_			84–87		[39,40]
	Carbamazepine	NF	PES	-	-	31–39	synthetic	[37]
			PA_TF_	-	-	>85		[42]
			PA_TF_	PVP	-	89		[43]
					Zeolite	85		
		RO	CA_TF_	-	-	85		[39]
			PA_TF_	-	-	91		
Lipid regulator	Clofibric acid	NF	PA_TF_	-	-	>85	synthetic	[42]
				PVP	-	>90		[43]
					Zeolite			
	Gemfibrozil	NF	PA_TF_	-	-	>85		[42]
				PVP	-	>90		[43]
					Zeolite			
		RO	PA_TF_	-	-	99.5		[41]
Non-steroidal anti-inflammatory	Acetaminophen	UF	PSU	MC	-	7	synthetic	[46]
					PAC	41.57		
			PI	-	-	15		[48]
					SiO_2_	99.9		[48]
		NF	PA_TF_	-	-	46		[44]
					ZIF-8	>55		
	Diclofenac	UF	PSU	MC	-	44.41	synthetic	[46]
					PAC	50.44		[46]
		NF	PES	-	-	55–61		[37]
			PA_TF_	-	-	85–93		[40,42]
				PVP	-	>90		[43]
					Zeolite			
		RO	PA_TF_	-	-	95		[40]
	Ibuprofen	UF	PI	-	-	11	synthetic	[48]
					SiO_2_	87		
		NF	PES	-	-	55–61		[37]
			PA_TF_			>85		[42]
		RO	PA_TF_	-	-	99.8		[41]
	Naproxen	NF	PA_TF_	-	-	>85	synthetic	[42]
	Phenacetine	NF	PA_TF_	-	-	19	synthetic	[40]
		RO	CA_TF_	-	-	10		[39]
			PA_TF_			71–74		[39,40]
Hormones and endocrine disruptive compounds.	17β-Estradiol	UF	PES	PVP	HMCN	94		[47]
		RO	CA_TF_	-	-	29	synthetic	[39]
			PA_TF_			83		
	Bisphenol A	UF	PES	PVP	-	25	water treatment plant	[60]
					SiO_2_	87		
			PVC	PVP	-	>40	synthetic	[59]
					COOH-MWCNT	>50		
					MWCNT/Fe_3_O_4_	57.4		
			PVDF		β-CDP	>99.9		[58]
TF—thin-film membrane								

### 2.2. Pesticide Removal

Pesticides are potential contaminants of drinking water supplies as they are applied to agricultural land, gardens, and lawns and can enter groundwater or surface water systems. Since they are in most cases very toxic to living beings and as human exposure to these compounds is high, EPA implemented regulations to protect the nation’s drinking water from the source to the tap. Thus, membrane filtration processes have also been considered for the removal of these types of pollutants, although to a less extent than PhACs. In this case, the use of nanomaterials was restricted only to the preparation of ultrafiltration membranes that include a novel cross-linked β-cyclodextrin polymer (β-CDP) with a hierarchically micro-mesoporous structure and high surface area as an additive. This membrane was preferentially tested for the separation of organic micropollutants, including 2,4-dichlorophenol [58]. According to the authors, the high removal efficiency of 2,4-dichlorophenol is related to the synergistic effect of the micropores and mesopores of β-CDP incorporated. The micropores offer a high number of adsorption sites that enhance the adsorption capacity, while the mesopores provide an unrestricted diffusion pathway and facilitate rapid mass transfer to achieve a high adsorption rate. Nanofiltration membranes have also been proposed for pesticide removal from water [62,63,64] (Table 2). Commercial membranes consisting of poly(vinyl alcohol)/polyamide [62,63] or TFC polyamide membranes prepared by interfacial polymerization of 1,3-phenylenediamine and 1,3,5 trimesoyl chloride coated on asymmetric polysulfone support [64] were used. Similar to PhACs, the studies show that the rejection % depends on the molecular weight, molecular width, and hydrophobicity of the pesticide [62,63,64], the steric hindrance being another important factor for solute permeation, even for hydrophobic pesticides. Overall, the highest average rejection efficiency was for persistent organochlorine insecticides (93%), including endosulfan (100%), DDT (95%), and HCH (92%). These results show a correlation between the rejection (%) of pesticides with their log P and molecular weight, according to Mukherjee et al. [64]. Strongly hydrophobic pesticides (log P > 4.5) such as DDT, bifenthrin, aldrin, permethrin, α-cypermethrin, ethion, difenoconazole, α-endosulfan, chlorpyrifos, and butachlor showed high rejection rates (80–100%), while less hydrophobic ones such as dimethoate (log P = 0.7) and thiamethoxam (log P = −0.13) showed lower removals (<80%); remarkably, low rejection (38%) was observed for monocrotophos (log P = −0.22). On the other hand, some pesticides with a molecular weight > 400, such as the isomers endosulfan and difenoconazole, were 100% rejected, while other pesticides with similar hydrophobicity (Log P > 3.8) but with a lower molecular weight had a lower rejection rate. For example, some organochlorine pesticides such as α-HCH, β-HCH, β-HCH, and dicofol were rejected at 89.18, 90.41, 88.18, and 72.17%, respectively.

In another work, similar conclusions were written by Kiso et al. [62,63], who used poly (vinyl alcohol)/polyamide membranes to remove aromatic [63] and non-phenylic pesticides [62] (Table 2). Almost all aromatic pesticides, except tricyclazole (79.6%) with the lowest molecular weight, were rejected (>92%) by the membrane confirming the molecular size exclusion effects. However, other pesticides with similar molecular weight as tricyclazole (PM = 207.3 g/mol; log P = 1.70) but with higher hydrophobicity, such as fenobucarb (PM = 189.2 g/mol; log P = 2.78) and carbaryl (PM = 201.2 g/mol; log P = 2.36) were rejected in higher proportion, showing that the hydrophobicity of compounds determines the rejection of aromatic pesticides [63]. On the other hand, all non-phenyl pesticides were removed by more than 96.7%, except dichlorvos, whose rejection was 86.7% [62]. As observed for aromatic pesticides, there is a synergistic effect between the low hydrophobicity (Log P ˂ 1.5) and small size (221 g/mol) of the compounds to be separated. The results shown in Table 2 of both studies indicate that the effects of the phenyl group were not significant, since the rejection of aromatic pesticides was only slightly lower than that of other pesticides.

### 2.3. Microorganism Removal

The removal or inactivation of bacteria from the drinking water is essential for the health and well-being of the population. Many water treatment systems use chemicals to kill or inactivate bacteria, but they can also be physically removed by membrane filtration (see Table 3). However, from Table 3 we can conclude that the MF and UF processes are not able to remove completely all the microorganisms even though the MW of microorganisms is higher than the MWCO of MF and UF membranes. In other words, the size exclusion removal mechanism seems to be not sufficient to completely remove the microorganisms. To achieve better removal, some authors propose the incorporation of antimicrobial nanoparticles in the membranes [65,66]. Silver oxide nano-sized particles have received special attention in this context, due to the strong bactericidal activity of this material. Coating the surface of a PA microfiltration membrane with silver oxide (AgO) Kacprzyńska-Gołacka et al. [66] observed a complete elimination of Gram-negative (*Escherichia coli*) and Gram-positive bacteria (Bacillus subtilis) from water (Table 3). According to the authors, the strong antibacterial properties of AgO-modified membranes may be associated with the release of silver ions and their ability to anchor and penetrate the external structures of the bacteria, causing damage to the cell membrane permeability, resulting in the death of the microorganisms. The free radical formation is also one of the arguments to explain cell death as a result of silver oxide nanoparticles leaching (Figure 2) [67].

PSU membranes impregnated with silver nanoparticles (nAg) [68] were shown to be effective against the *E. coli* K12 strain of bacteria, due to the release of Ag+ ions that influences the antimicrobial activity. The presence of this nanomaterial improves the effectiveness of the membrane by preventing bacteria adhesion to the membrane surface and reducing biofilm formation. According to them, the interaction of the silver cation with thiol groups and the formation of S-Ag bonds or disulfide bonds can destroy bacterial and viral proteins, interrupting the electron transport chain and interacting with DNA. Nanomaterials with strong photocatalytic properties, such as titanium dioxide (TiO_2_), have been shown to improve the efficiency of bacteria removal through the photocatalytic production of reactive oxygen species that damage cell components and viruses [69]. Despite its chemical stability, low toxicity, low pollutant load, and availability at low cost, its use has certain limitations. TiO_2_ shows bactericidal activity only in the presence of UV radiation. Therefore, membrane modifications with TiO_2_ + AgO have been proposed to improve photocatalytic effect in the visible region [65]. Removal rates of 100% were achieved since both nanoparticles were used and associated with magnetron sputtering technology, helping to create new structural properties in the polymeric membranes.

**Table 3 membranes-12-00570-t003:** Application of polymeric membranes in the purification of water contaminated with microorganisms.

Micro-Organisms	Process	Polymer	Additive	Nanomaterial	% Removal	Sample	Ref.
Bacillus subtilis	MF	PA	-	-	0	synthetic	[66]
				AgO	100		
				TiO_2_-AgO	100		[65]
Bacteriophage MS2	MF	PVDF	-	-	32		[70]
	UF	PSU	PVP	nAg	100		[68]
*Escherichia Coli*	MF	PA	-	-	0		[66]
				AgO	100		
				TiO_2_-AgO	100		[65]
		PVDF	-	-	42		[69]
				TiO_2_	100		
	UF	PES	-	-	no clear		[71]
				nAg	99.99		
		PSU	PVP	-	50		[68]
				nAg	99		

### 2.4. Dye Removal

The dye and textile dyeing industry is the main industry responsible for the contamination of surface water and groundwater. The dyes significantly compromise the aesthetic quality of water bodies, increase biochemical and chemical oxygen demand (BOD and COD), impair photosynthesis, inhibit plant growth, enter the food chain, provide recalcitrance and bioaccumulation, and may promote toxicity, mutagenicity, and carcinogenicity. Membrane filtration processes have been an alternative to the common remediation processes for these pollutants. Generally, the application of UF processes for the removal of dissolved solutes in aqueous media is not very effective. However, it has been shown that in the case of dyes they can achieve removal rates between 70 to 82% (Table 4). This may be related to the high molecular weight of most dyes, which is higher than the MWCO of these membranes. In the case of NF membranes, as mentioned above, they offer higher organic contaminant rejection than UF membranes, with dye removal rates of around 90%. Several types of membrane polymers have been studied for dye removal, among which the most investigated are PES, followed by PSU. As shown in Table 4, PES membranes without the addition of nanomaterials present a high removal rate of about 89, 90, and 93.2% for the direct yellow 12 (DY12) [72], direct red 16 (DR16) [73], and reactive green 19 (RG19) [72] dyes, respectively, except for dye RB21 [72], which was only removed at 61.4%. On other hand, PSU membranes without nanomaterials were evaluated for eosin yellow [74] with a remotion rate of about 67%. This removal rate was lower than DY12 (89%) mentioned in the previous study from Safarpour et al. [72], both with a similar molecular weight. This result can be explained by the different molecular weight cutoff of membranes used.

The modification of the membranes by the addition of nanoparticles, such as aluminium oxide (Al_2_O_3_), graphene oxide (GO), TiO_2_ or GO/TiO_2,_ and nanoscale zero-valent iron (nZVI), have been studied for different dyes removal, as shown in Table 4 [72,73,74,75]. According to the data, PES membranes enriched with GO improve the removal rate by more than 8% for the DR16 and RB21 dye, but no significant effect was observed for other studied dyes [72,73]. These results may be related to the presence of the GO nanoparticles that can induce a negative charge on the surface of the nanostructured membrane and give rise to an electrostatic repulsion effect with the dye molecules that have a negative charge at neutral pH (Donnan exclusion mechanism). Similarly, TiO_2_ also produces better removal rates for some of the studied dyes. The remotion rate of RB21 dye increases around 10% for PSU membranes containing TiO_2_ when compared to the same polymer membrane without any nanoparticle [72]. Moreover, a remotion rate improvement of 30% for eosin yellow dye was observed for the PSU membrane enriched with TiO_2_ [74]. These results are attributed to the increased porosity of the membranes embedded with nanoparticles, which allows faster penetration of the dye into the membrane. These conditions favor adsorption of the dye onto the photocatalyst, resulting in higher removal rates. Safarpour et al. [72] reported the better dye rejection properties of the combination of nanoparticles GO/TiO_2_ in PES membranes when compared with individual use. These can be attributed to the synergic action of both nanomaterials. The presence of GO can decrease aggregation and improve the photocatalytic efficiency of TiO_2,_ because it has several acidic functional groups on its surface, which facilitates the anchoring of TiO_2_ and thus establishes a longer and closer contact between TiO_2_ and the dyes. In addition, these GO/TiO_2_-modified membranes showed better overall properties, such as water permeability and fouling resistance compared with those modified with GO and TiO_2_ separately. Recently, Rajeswari et al. [75] proposed an ultrafiltration CA-PSU membrane incorporating nZVI and Al_2_O_3_ for the remotion of methylene blue (MB). Metal nanoparticles proved to be suitable for improving the removal efficiency of the membranes, with a remotion rate of 91 and 94% for the membrane incorporating Al_2_O_3_ and nanoscale zero-valent iron (nZVI), respectively. These results were superior to those obtained by Safarpour et al. [72] for a higher molecular weight dye such as RB21 with a TiO_2_-containing PES membrane and a higher molecular weight cut-off. Therefore, Al_2_O_3_ y nZVI appear to be more promising than TiO_2_ for dye removal.

### 2.5. Heavy Metals Removal

The main threats of heavy metals to human health are associated with exposure to lead, cadmium, mercury, and arsenic. These metals have been extensively studied and international bodies such as WHO have regularly reviewed their effects on human health. These inorganic pollutants are being discharged in waters, soils, and into the atmosphere due to the rapidly growing agriculture and metal industries, improper waste disposal, fertilizers, and pesticides. The use of membranes for filtration has also been selected as a remediation process for these types of contaminants, mainly by UF and NF for heavy metals removal (see Table 5). Similar to microorganisms, the size exclusion removal mechanism seems to be not sufficient to efficiently remove heavy metals from water and even shows poor removal rates (between 10 to 35%). It has been shown that commercial TFC membranes and conventional membranes incorporated with some nanomaterials can improve the removal efficiency of these processes. Commercial polyamide TFC nanofiltration membranes have achieved high metal removal from synthetic wastewater at levels of >95% [77,78]. The authors have shown that the metal retention percentage is mainly pH-dependent, since the charge property of the membrane surface material changes with the pH. These results provide a basis for the application of such membranes in the separation of metals from wastewater.

Carbon-based nanomaterials have been shown to have the potential to selectively remove heavy metal ions from water sources [79,80]. Carbon nanotubes increase heavy metals removal because they improve the adsorption capacity of the membranes and reduce the size of the pores in the range of 20 to 30 nm. Small pore sizes and high adsorption help make membranes efficient enough to remove metals from water, either by size exclusion mechanisms, by adsorption, or by a combination of both [79]. Although carbon nanotubes have excellent properties to improve the performance of membranes, in some cases their functionalization is recommended to help their dispersion and better performance [79,81]. Different PSU membranes incorporating functionalized azide-MWCNT, amide-MWCNT, and oxidized-MWCNT were evaluated [79]. Higher removal rates for Cd (II), Cr (VI), Cu (II), and Pb (II) were obtained when azide-MWCNT and amide-MWCNT were incorporated against oxidized-MWCNT, except for arsenium ion (Table 5). The better performance is justified by the presence of amide functionalized (-CONH and CH2N2) or azide-functionalized (-CON3) groups which have higher complexation constants with metal ions than the -COOH and -OH groups of the oxidized MWCNTs. It should be noted that these sites for complexation with metal ions are absent in pure PSU membranes, explaining their low efficiency (~10%) in heavy metals removal. On the contrary, arsenic is better removed with oxidized-MWCNTs (83.6%) [79] and GO (83.65%) [80] than with amide-MWCNTs (79.4%) when a PSU membrane is used. This result was expected due to the high ability of the -COOH and -OH groups to repel As [79,80]. As mentioned above, the -COOH and -OH groups on the GO and oxidized-MWCNT surface can induce negative charges on the membrane surface, which causes electrostatic repulsion of the ions resulting in increased rejection of the As. In addition to the previous results, Vinodhini et al. [82] observed that CA membranes modified with PEG and nanochitosan were also very effective for Cr(VI) removal (95%). This result may also be related to the positive charge of the membrane due to the incorporation of chitosan nanoparticles in its matrix, which would improve its adsorption capacity. For all the described heavy metals, only cadmium presents remotion rates lower than 80%, independently of any strategy adopted to modify the membranes (Table 5).

Metal oxide nanoparticles are another group of important nanomaterials with a large surface area, high activity, high adsorption capacity, and selectivity. Different metal oxide nanoparticles have been compared concerning the remotion of copper in PES membranes [83,84,85]. The presence of AL_2_O_3_ nanoparticles in pristine membranes improves the removal rate from 25% to 60% [85], contrasted by the use of Fe_3_O_4_ that only increases 5% [84]. This was expected, since the high affinity of AL_2_O_3_ for heavy metals is described [85]. In addition, AL_2_O_3_ nanostructured membranes had better water flux compared with pure PES membranes due to the higher hydrophilicity provided by the nanoparticles. To ameliorate the obtained results, Ghaemi et al. [84] coated and functionalized the Fe_3_O_4_ nanoparticles with more hydrophilic materials such as silica, metformin-modified silica, and amine-modified silica that enable better nanoparticle dispersion into the membrane. This strategy favors a better elimination of Cu (II) with acceptable levels of permeability, being the Fe_3_O_4_ nanoparticles coated with metformin-modified silica, the best combination to remove around 92% of Cu (II). These results are explained by the large number of N atoms around each particle that offer more available sites for adsorption on the membrane surface. In addition, the hydrophilicity of the modified nanoparticles may increase the available adsorption sites and thus the migration of heavy metals to the membrane surface. This approach was more efficient than the approach obtained by Daraei et al. [83], where Fe_3_O_4_ nanoparticles were modified with polyaniline, a polymeric material usually applied as a modifier to achieve greater adsorption of heavy metals on nanoparticles. The authors observed the improvement in copper adsorption capacity by the existence of NH groups of polyaniline as reactive sites for adsorption. However, they noticed a reduction in the water flow due to the clogging of the membranes by the nanoparticles. As stated above, a higher rejection % of copper was reported for PSU membranes incorporating azide-MWCNTs [79]. These results show that carbon-based nanomaterials are more efficient in Cu removal when compared with metal oxide nanoparticles. However, studies using a CA-PSU membrane [75] modified with Al_2_O_3_ or nZVI showed no significant differences with both membranes in copper removal (≈84–88%); nevertheless, the rejection rate was higher than the membrane without any nanoparticles (78%).

**Table 5 membranes-12-00570-t005:** Application of polymeric membranes in the purification of water contaminated with heavy metals.

Heavy Metals	Process	Polymer	Additive	Nanomaterial	% Removal	Sample	Ref.
Arsenic (As)	UF	PSU	-	-	10.9	synthetic	[79]
				Amide-MWCNT	79.4		
				Azide -MWCNT	80.9		
				Oxidized- MWCNT	83.6		
				GO	83.65		[80]
Cadmium (Cd)	UF	PSU	-	-	9.9		[79]
				Amide-MWCNT	78.2		
				Azide -MWCNT	79.1		
				Oxidized- MWCNT	71.6		
Chromium (Cr)	UF	CA	-	-	35.72	synthetic	[86]
			PEG	-	31.89		
				nanochitosan	95	Tannery effluent	[82]
		PSU	-	-	10.2	synthetic	[79]
				Amide-MWCNT	94.2		
				Azide -MWCNT	94.8		
				Oxidized- MWCNT	86.2		
	NF	PA_TF_	-	-	96–99		[77,78]
Copper (Cu)	UF	CA	PVP	-	29	synthetic	[87]
		CA/PSU	-	-	78	wastewater	[75]
				Al_2_O_3_	84		
				nZVI	88		
		PSU	-	-	10.1	synthetic	[79]
				Amide-MWCNT	93.1		
				Azide -MWCNT	93.9		
				Oxidized- MWCNT	79.3		
	NF	PES	PVP	-	25		[85]
				AL_2_O_3_	60		
				Fe_3_O_4_	∼30		[84]
				Fe_3_O_4_/SiO_2_	∼40		
				Fe_3_O_4_/SiO_2_-Met	∼92		
				Fe_3_O_4_/SiO_2_-Amide	∼75		
				PANI/Fe_3_O_4_	80–85		[83]
		PA_TF_	-	-	95.33		[78]
Lead (Pb)	UF	PSU	-	-	10.5	synthetic	[79]
				Amide-MWCNT	90.1		
				Azide -MWCNT	90.8		
				Oxidized- MWCNT	41.3		
Nickel (Ni)	NF	PA_TF_	-	-	94.99	synthetic	[78]
TF—thin-film membrane							

### 2.6. Mycotoxins Removal

Mycotoxins are a group of metabolites produced by fungi that have received special attention in recent years because they are highly toxic and carcinogenic to animals and humans [88]. Although mycotoxins represent a major risk to human health, as far as the author’s knowledge, there are no reports of drinking water contaminated with mycotoxins, which is reflected in the limited interest in the removal of these contaminants. However, due to their constant presence in crops and food, there is potential contamination of drinking water with these compounds. Therefore, only two works in the literature refer to the use of filter membranes for the elimination of these contaminants by UF (Table 6). These low-pressure membranes have been shown to be inefficient in removing mycotoxins from water (Table 6). However, their functionalization with MOFs can improve the removal efficiency of these processes [89].

MOFs have been used for sensing applications, catalysis, drug distribution, separation, and also as adsorbents for the removal of some contaminants from aqueous systems [90]. However, the functionalization of filtration membranes with such compounds is still scarce. Only two articles were found related to the use of functionalized filtration membranes with MOFs for mycotoxin removal (Table 6). Ren et al. [89] prepared three types of polyacrylonitrile (PAN) membranes nanostructured with different Fe-based MOFs (MIL-100, MIL-53, and MIL-68) to mimic the enzyme peroxidase and to withhold aqueous mycotoxins (aflatoxin B1) from synthetic water samples. The prepared membranes simultaneously had the ability to filter, adsorb, and catalyze the mycotoxins understudy by the presence of MOFs in their composition. The removal efficiency for aflatoxin B1 (AFB1) was 74.9, 20, and 10% when MIL-100, MIL-53, and MIL-68 were used, respectively. Differences in MOF structure and Fe ion active sites are responsible for the results obtained. Thus, the results may be related to the short distance that MIL-68 has between the terephthalic acid connectors along the pore axis, so it cannot form a mutually permeable structure, resulting in decreased pore volume. In addition, the small size of its triangular channels may make it difficult for toxins to enter the triangular pores, which explains the lower removal efficiency of MIL-68 membranes. Although MIL-68 and MIL-53 have the same organic linker, water molecules can form hydrogen bonds with MIL-53 through the hydrophilic inorganic part in the pores. As a result, the pore size could increase to about 13 Å, so AFB1 (10.8 Å × 8.7 Å) could enter the pores and be adsorbed by MIL-53 through the electrostatic interaction and π-π interaction between the organic structure and the aromatic ring of AFB1. Due to the stronger interaction between MIL-100 and water molecule compared with MIL-53 (resulting in more severe competitive adsorption), the adsorption capacity of AFB1 was higher with MIL-100 [91]. Although the results are promising, especially for MIL-100, the removal rates are still low. However, the same authors demonstrated that for the three membranes described, the adsorption capacity of Aflatoxin B1 improved when H_2_O_2_ was used to degrade the molecules attached to the MOF-loaded membrane. Therefore, increasing the catalytic activity of the MOFs, by adding H_2_O_2_, prevents the adsorption of the toxin to the membrane, reducing its clogging and increasing the removal rate.

**Table 6 membranes-12-00570-t006:** Application of polymeric membranes in the purification of water contaminated with mycotoxins.

Mycotoxins	Process	Polymer	Nanomaterial	% Removal	Sample	Ref.
Aflatoxin B1	UF	PAN	-	>10	synthetic	[89,91]
			MOF (MIL-100)	70–74.9		
			MOF (MIL-53)	>20		
			MOF (MIL-68)	10		

### 2.7. Policyclic Aromatic Hydrocarbon (PAH) and Phthalate (PAE) Removal

There are several sources of this type of contaminant that result from natural activities (i.e., forest fires and volcanic activity), domestic, industrial, agricultural, and even mobile sources (i.e., aircrafts exhaust, oil tanks/ships), etc. They persist in the environment and accumulate in biota and food chains and have potential adverse effects on aquatic life and humans, including carcinogenic properties. Therefore, the contamination of water resources by alkyl phthalates and some hazardous phenyl compounds has also been under the attention of researchers. There are few works in the literature that focus on the use of filter membranes for these types of compounds [56,92] (Table 7). Commercial nanofiltration membranes based on sulfonated polyethersulfone or polyvinylalcohol/polyamide for alkyl phthalates and other solutes such as mono-substituted benzenes have been used with relative success [92]. The results showed that the latter has rejection rates higher than 99% for almost alkyl phthalates. However, p-dimethyl phthalate, p-diethyl phthalate, and all monosubstituted benzenes presented significantly lower rejection rates. This may be related to the fact that these compounds had a smaller molecular size when compared with the compounds mentioned above (<0.32 nm). On the other hand, strongly hydrophobic alkyl phthalates (Log P > 5) such as di-n-octyl phthalate and di-(2-ethylhexyl) phthalate were almost entirely rejected. Therefore, hydrophobic interaction between the solute and the membrane is an important factor in the permeation of these kinds of compounds. As observed for pesticides, the rejection of alkyl phthalates is also influenced by the molecular size and hydrophobicity of the molecules.

The functionalization of polysulfone membranes with β-cyclodextrin (β -CD), as shown by Choi et al. [56], leads to a maximum removal rate of 70% for di-(2-ethylhexyl) phthalate in drinking water. Although this result is less relevant than the one obtained by TFC membrane (99%) previously described, it nevertheless allows for higher flux and lower operating pressures.

## 3. Conclusions

The elimination of various contaminants in water can be carried out by a physical process through filtration membranes. Several polymeric membranes have been developed for PhAC, pesticide, microorganism, dye, heavy metals, mycotoxin, PAH, and PAE rejection in water. The use of various types of polymers modified by additives and/or nanomaterials have been reported in the literature. From the systematic literature review some conclusions are highlighted:

The type of polymer determines the physicochemical characteristics and the performance of the membranes. Therefore, the polymer choice should be based on the purposed application.

Charged membranes give rise to an electrostatic effect with charged molecules (Donnan exclusion mechanism), being important to control the pH of feed solutions, although it is not possible to establish any correlation because there is no experimental evidence.

Conventional polymeric membranes have low efficiency for removing contaminants from water, except for dyes, where removal rates are about 90%.

Thin-film composite (TFC) membranes have better removal capacity than conventional membranes, although with higher operating pressures, which limits their use.

The incorporation of polymeric additives in membranes does not provide greater selectivity in the membranes; however, they can improve the dispersion of nanomaterials in nanostructured membranes, thus obtaining more effective incorporation.

The impregnation of nanomaterials into polymeric membranes is a promising alternative to overcome their limitations, i.e., fouling, surface area, hydrophilic properties, taking advantage of their functional properties.

According to the target contaminant to be removed from the water, different nanomaterials were considered. For the removal of microorganisms, silver nanoparticles are the most suitable due to their high antimicrobial activity; the combination of GO/TiO_2_ nanoparticles is more effective than their individual use for dye removal; for heavy metals removal, carbon nanotubes and certain metal nanoparticles are the most efficient and widely used. In the case of carbon nanotubes, their functionalization improves membrane ability to remove metal ions compared to non-functionalized nanotubes. A new generation of adsorbent materials such as MOFs are effective in removing emerging contaminants such as mycotoxins and some PhACs. Other nanomaterials, such as nanosilica (SiO_2_) and some macromolecules (cyclodextrins), are also quite effective in the rejection of certain PhACs. However, more research is needed to prove their ability for a larger number of contaminants.

Thin-film nanostructured (TFN) membranes exhibit higher fluxes compared with the highly cross-linked non-porous polyimide layers typical of TFC membranes. Therefore, they could be a promising alternative to achieving a more effective separation of contaminants at lower energy costs.

## Figures and Tables

**Figure 1 membranes-12-00570-f001:**
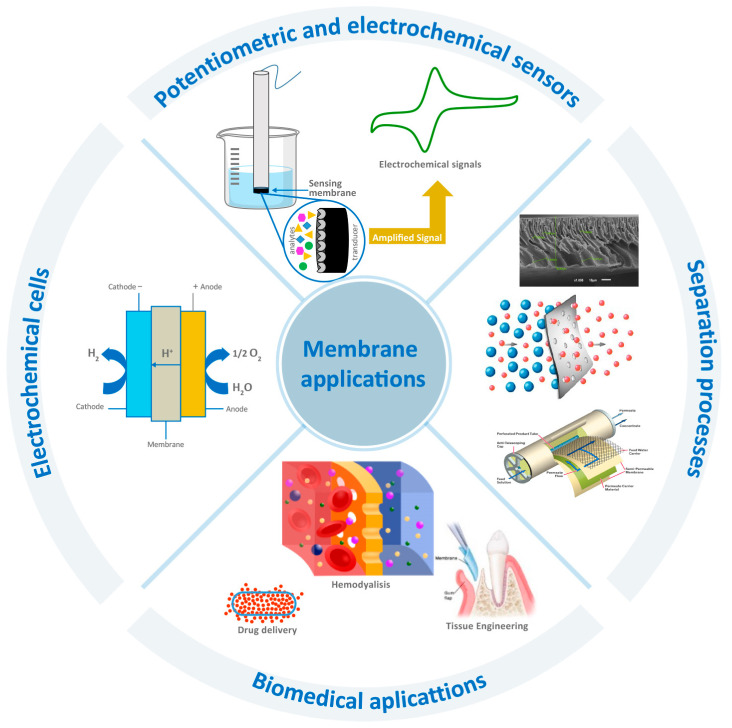
Application of polymeric membranes.

**Figure 2 membranes-12-00570-f002:**
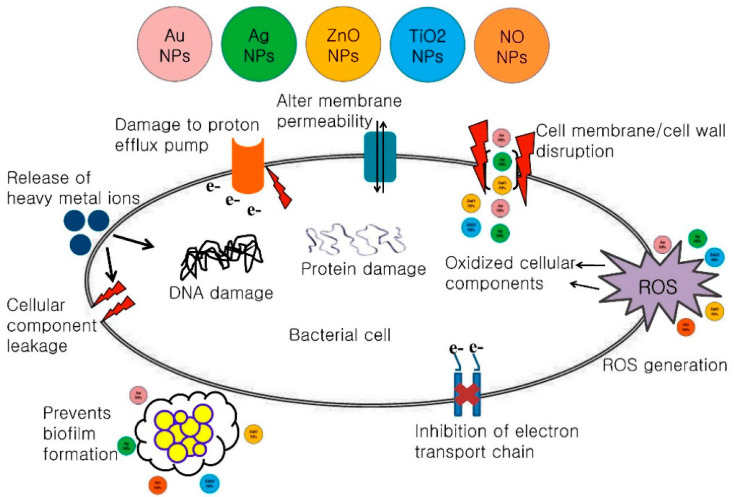
Schematic representations of the antimicrobial mechanisms of various nanoparticles (NPs) [67].

**Table 2 membranes-12-00570-t002:** Application of polymeric membranes in the purification of water contaminated with pesticides.

Pesticide Class	Name	Process	Polymer	Additive	Nanomaterial	% Removal	Sample	Ref.
Benzimidazole fungicide Triazole fungicides	Carbendazim	NF	PA_TF_	-	-	64.15	synthetic	[64]
	Difenoconazole					100		
	Hexaconazole					79.38		
	Propiconazole		PVA/PA			96.9		[63]
	Tetraconazole		PA_TF_			72.94		[64]
Carbamate insecticides	Carbaryl	NF	PVA/PA	-	-	86–92	synthetic	[63,64]
	Carbofuran		PA_TF_			89.98		[64]
	Esprocarb		PVA/PA			99.94		[63]
	Fenobucarb					94.8		
	Thiram					97.7		[62]
	Molinate					98.5		
Chloroacetamide herbicides	Alachlor	NF	PA_TF_	-	-	86.18	synthetic	[64]
	Butachor					100		
Chlorophenoxy herbicide derivative	2,4-dichlorophenol	UF	PVDF	PVP	β-CDP	99.9	synthetic	[58]
Neo-nicotinoid insecticides	Acetamiprid	NF	PA_TF_	-	-	81.05	synthetic	[64]
	Imidacloprid					89.17		
			PVA/PA			97.6		[62]
	Thiachloprid		PA_TF_			80.58		[64]
	Thiamethoxam					66.61		
Organochlorine insecticides	Aldrin	NF	PA_TF_	-	-	89.61	synthetic	[64]
	α-Endosufan					100		
	α-HCH					89.18		
	β-Endosulfan					100		
	β-HCH					90.41		
	δ-HCH					88.18		
	Dicofol					72.17		
	Dieldrin					82.56		
	Endosulfan sulphate					100		
	γ-HCH					99.85		
	op-DDD					94.47		
	op-DDE					95.07		
	op-DDT					94.64		
	pp-DDD					94.13		
	pp-DDE					95.95		
	pp-DDT					96.02		
Organophosphorus insecticides	Chlorpyrifos	NF	PA_TF_	-	-	86.9	synthetic	[64]
			PVA/PA			>99.9		[62]
	Diazinon					99.52		
	Dimethoate		PA_TF_			73.67		[64]
	Dichlorvos		PVA/PA			86.7		[62]
	Isoxathion					99.84		[63]
	Ethion		PA_TF_			90.94		[64]
	Malathion					55.51		
			PVA/PA			99.64		[62]
	Methyl parathion		PA_TF_			48.26		[64]
	Monocrotophos					37.82		
	Parathion					55.61		
Phenyl-amide fungicide	Metalaxyl	NF	PA_TF_	-	-	85.64	synthetic	[64]
Phosphorothiolate fungicide	Isoprothiolane	NF	PA_TF_	-	-	85.49	synthetic	[64]
			PVA/PA			99.76		[62]
Synthetic pyrithroid insecticides	α-Cypermethrin	NF	PA_TF_	-	-	84.27	synthetic	[64]
	Bifenthrin					87.26		
	Permethrin					80.14		
Thiazole fungicide	Mefenacet	NF	PVA/PA	-	-	99.1	synthetic	[63]
	Tricyclazole		PA_TF_			81.05		[64]
			PVA/PA			79.6		[63]
Triazine herbicide	Atrazine	NF	PVA/PA	-	-	93–97.5	synthetic	[62,64]
	Simazine					96.7		[62]
	Simetryn					98.6		
Urea herbicide	Isoproturon	NF	PA_TF_	-	-	87.25	synthetic	[64]
TF—thin-film membrane								

**Table 4 membranes-12-00570-t004:** Application of polymeric membranes in the purification of water contaminated with dyes.

Dyes	Process	Polymer	Additive	Nanomaterial	% Removal	Sample	Ref.
methylene blue (MB)	UF	CA/PSU	-	-	82	wastewater	[75]
				Al_2_O_3_	91	wastewater	
				nZVI	94		
eosin yellow	UF	PSU	-	-	67	synthetic	[74,76]
				TiO_2_	87–97		
direct red 16 (DR16)	NF	PES	PVP	-	90		[73]
				GO	99		
direct yellow 12 (DY12)	NF	PES	PVP	-	89		[72]
				GO	>90		
				TiO_2_	>90		
				GO-TiO_2_	95.4		
reactive green 19 (RG19)	NF	PES	PVP	-	93.2		
				GO	>90		
				TiO_2_	>90		
				GO-TiO_2_	99.4		
reactive blue 21 (RB21)	NF	PES	PVP	-	61.4		
				GO	69.7		
				TiO_2_	73.5		
				GO-TiO_2_	81.4		

**Table 7 membranes-12-00570-t007:** Application of polymeric membranes in the purification of water contaminated with polycyclic Aromatic Hydrocarbons (PAHs) and Phthalates (PAEs).

Hydrocarbons/ Phthalates	Process	Polymer	Additive	Nanomaterial	% Removal	Sample	Ref.
Aniline	NF	PVA/PA	-	-	17.9	synthetic	[92]
Anisole					27.8		
Benzene					62.0		
Chlorobenzene					63.4		
Dimethyl phthalate					96.4		
p-Dimethyl phthalate					65.1		
Diethyl phthalate					98.4		
p-Diethyl phthalate					80.5		
Di-n-propyl phthalate					99.6		
Di-iso-propyl phthalate					99.1		
Di-n-butyl phthalate					99.4		
Di-iso-butyl phthalate					99.8		
Dicyclohexyl phthalate					99.8		
Di-n-octyl phthalate					≧99.9		
Nitrobenzene					50.6		
Toluene					66.9		
Phenol					23.4		
Di-(2-ethylhexyl) phthalate					99.9		
	MF-UF	PSU	PVP	β-CD	70		[56]

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
