# Peer review of "Removal of Contaminants from Water by Membrane Filtration: A Review"

_membranes, 2022, doi:10.3390/membranes12060570_

Round 1

Reviewer 1 Report

This paper organizes the current issues of membrane for purification of various wastewaters. The detailed lists of wastewater are classified into seven items. And they suggest the present membrane technologies based on materials and processes for purifying and removing contaminants. The selection of references for each item is good for unprofessional readers who could catch up easily. However, to provide more attractive things, the paper must describe the basic of membrane science. What is RO and NT, etc based on the pore size or free volume size? Thus, the various contaminants have different molecular weights. Based on the MW, the membrane process is determined, as you know well. Furthermore, if the contaminant has the charges, the feed conditions must be suggested because of the Donnan removal effect. Thus, when some persons might read the paper, the paper must help to image and design the next-generation membranes. Therefore, I suggest, could you add a physical explanation of membranes in each item? 

Author Response

Dear Reviewer,

We would like to thank the comments and recommendations, which were thoroughly accepted to improve the manuscript quality. The original manuscript was carefully revised and modified, whenever possible, to comply  all points raised by you.

The changes are highlighted in the manuscript below uploaded.

With my best regards

Reviewer 2 Report

In my opinion the manuscript is relevant and provides interesting information about the removal of contaminantes in water by membranes.

Author Response

Dear Reviewer,

Thank you very much for your comments.
